# Gaining New Insights into Fundamental Biological Pathways by Bacterial Toxin-Based Genetic Screens

**DOI:** 10.3390/bioengineering10080884

**Published:** 2023-07-25

**Authors:** Songhai Tian, Nini Zhou

**Affiliations:** 1State Key Laboratory of Natural and Biomimetic Drugs, Department of Molecular and Cellular Pharmacology, School of Pharmaceutical Sciences, Peking University, 38 Xueyuan Road, Beijing 100191, China; 2Department of Urology, Boston Children’s Hospital, Boston, MA 02115, USA; nini.zhou@childrens.harvard.edu; 3Department of Microbiology and Department of Surgery, Harvard Medical School, Boston, MA 02115, USA

**Keywords:** bacterial toxin, Shiga toxins, cholera toxin, ricin, large clostridial toxins, genetic screen, CRISPR-Cas9, glycosphingolipids, protein glycosylation, membrane vesicle trafficking

## Abstract

Genetic screen technology has been applied to study the mechanism of action of bacterial toxins—a special class of virulence factors that contribute to the pathogenesis caused by bacterial infections. These screens aim to identify host factors that directly or indirectly facilitate toxin intoxication. Additionally, specific properties of certain toxins, such as membrane interaction, retrograde trafficking, and carbohydrate binding, provide robust probes to comprehensively investigate the lipid biosynthesis, membrane vesicle transport, and glycosylation pathways, respectively. This review specifically focuses on recent representative toxin-based genetic screens that have identified new players involved in and provided new insights into fundamental biological pathways, such as glycosphingolipid biosynthesis, protein glycosylation, and membrane vesicle trafficking pathways. Functionally characterizing these newly identified factors not only expands our current understanding of toxin biology but also enables a deeper comprehension of fundamental biological questions. Consequently, it stimulates the development of new therapeutic approaches targeting both bacterial infectious diseases and genetic disorders with defects in these factors and pathways.

## 1. Introduction

Bacterial exotoxins are special virulence factors responsible for many infectious diseases caused by bacterial pathogens [1,2]. Once being produced and released, these toxins autonomously target host cells by recognizing cell surface receptors through their highly specific receptor-binding moieties. Some toxins, such as membrane-damaging toxins and pore-forming toxins, act directly on the plasma membrane. In contrast, the more dangerous toxins, including retrograde trafficking toxins and single-chain toxins, deliver their enzymatic moieties across the membrane to act on their intracellular substrates (Figure 1). Extensive discussions have focused on the identification, mechanisms, and significance of these toxins [3,4,5,6,7,8,9].

Throughout the intoxication process, toxins exploit multiple host machinery to facilitate their attachment, uptake, translocation, and activation (Figure 1). Identifying and characterizing these host factors and pathways can provide important comprehension of the basic aspects of host–toxin interactions and aid in discovering novel anti-toxin therapeutics. Moreover, some newly identified host factors (excluding receptors) do not physically interact with toxins. Instead, they may be involved in complex cellular processes that are indirectly utilized by the toxins. In this case, toxins can serve as valuable probes to elucidate the roles of these novel players in fundamental biological pathways.

A key task in genetic analysis is to connect a specific phenotype with a gene. Reverse genetic approaches firstly generate a precise genetic perturbation and then track the consequential phenotypes. In contrast, forward genetic screens firstly modulate a panel of genes and then identify the gene responsible for a phenotype of interest [10]. These screens have discovered fundamental biological pathways in model organisms such as yeast, worms, fish, rodents, and humans [11,12].

The comprehensive genome-wide forward genetic screen is a straightforward and unbiased strategy to uncover unknown host factors for a toxin. This is because the surviving cells after toxin treatment may carry mutations in specific genes that have been involved in the toxin-induced killing (Figure 2). Historically, gain-of-function screens were introduced to the toxin field earlier than loss-of-function approaches because the classic technologies, such as complementary DNA (cDNA) expression cloning [13,14], could efficiently lead to ectopic overexpression of genes in mammalian cells in a high-throughput manner. The cellular receptors of many important toxins, such as diphtheria toxin (DT), were identified using this strategy [15]. Recently, technological advancements such as RNA interference (RNAi) [16,17], retrovirus-based insertional mutagenesis in haploid cells [18,19], and the recently developed CRISPR (clustered regularly interspaced short palindromic repeat) system [20,21] have made loss-of-function screens more popular. In particular, the CRISPR system, consisting of CRISPR-associated Cas9 nuclease and single guide RNA (sgRNA), has revolutionized the field of genetic screen due to its ability to induce strong loss-of-function mutations (knockout) at precise loci on both alleles in diploid genomes [22,23]. As an alternative approach, the CRISPR repression (CRISPRi) system blocks the transcription and achieves mild loss-of-function (knockdown) [24,25]. On the other hand, the emerging gain-of-function screen relies on CRISPR activation (CRISPRa) technology [26], which is also valuable for studying host–toxin interactions (Figure 2).

The RNAi screen is easy to handle and can be applied to multiple cell lines to achieve high genome coverage. However, this post-transcriptional gene repression approach has significant off-target effects and may lead to hypomorphic mutations, where the targeted genes are partially suppressed [27]. Compared to RNAi, CRISPRi affects chromatin and results in more effective knockdown outcomes [26]. The retrovirus-based insertional screen offers a highly efficient method for generating genome-wide loss-of-function. However, it requires haploid or nearly haploid cells, and the insertion sites have a virus-dependent preference [28], which may result in incomplete genome coverage. In gain-of-function approaches, cDNA expression cloning is widely used, but the expression levels of each gene are not well controlled, leading to inconsistencies in phenotypic outcomes [14]. In contrast, the CRISPRa approach offers a more controlled method to specifically activate gene expression [29]. It should be noted that all the Cas9/sgRNA-based mutagenesis strategies may have a certain level of off-target effects [30].

The genome-wide CRISPR-Cas9-mediated loss-of-function screen is the most widely used strategy with multiple advantages. Notably, although complete knockout mutations can be generated at most alleles, some cells may retain a copy of the allele with partial function generated by non-frameshift mutations [22]. A typical screen begins through generating a knockout cell library by introducing a sgRNA library (e.g., the GeCKO-v2 library [20]) into the Cas9-expressing cells. Subsequently, the cells are exposed to toxins as a phenotypic selection, usually through multiple rounds, to minimize contaminants. The surviving cells at the end of the selection can be expanded and analyzed using advanced sequencing techniques to determine the specific genes involved (Figure 2) [31,32,33,34,35].

This review specifically focuses on the recent representative genetic screens that relied on classic bacterial toxins (e.g., Shiga toxins, cholera toxin, ricin, diphtheria toxin, *Pseudomonas* exotoxin A, anthrax toxin, large clostridial toxins, and pore-forming toxins) as screening stresses. These screens identified and characterized host factors that indirectly interact with toxins but play important roles in fundamental biological pathways such as glycosphingolipid biosynthesis, protein glycosylation, membrane vesicle trafficking, and other unique pathways. The objective of this review is to inspire innovative approaches that utilize toxin-based platforms to make fundamental breakthroughs on basic biological questions beyond a deep understanding of the toxin’s mechanism of action.

## 2. Factors Required for the Biosynthesis of Glycosphingolipids

As integral components of the cell membrane, glycosphingolipids (GSLs) consist of a glycosidically bound carbohydrate moiety and a lipid moiety known as ceramide (Figure 3). The carbohydrate moieties of GSLs can interact with other carbohydrates or proteins, serving as the molecular basis of cell–cell recognition and initiating cellular activities such as immune response, cell proliferation, and apoptosis [36,37]. Additionally, GSL-organized microdomains on the plasma membrane provide a molecular platform for clustering the proteins involved in signal transduction [36,37]. Although GSLs are essential for tissue development, excessive accumulation of GSLs can cause a class of inherited disorders called sphingolipidoses (e.g., Fabry disease, Gaucher disease, and Niemann–Pick disease) [38,39]. Therefore, a deep understanding of the GSL metabolism pathways, encompassing both biosynthesis and degradation, holds promise for developing therapeutic approaches.

The de novo biosynthesis of GSLs (Figure 3) in mammals starts with ceramide, which is synthesized on the endoplasmic reticulum (ER) membrane through four steps of biosynthesis, originally from serine and palmitoyl-CoA. Ceramide is then transported to the Golgi apparatus, where a series of glycosyltransferases catalyze the transfer reactions of carbohydrate moieties between donor and acceptor molecules [41]. UGCG (UDP-glucose ceramide glucosyltransferase) transfers UDP-glucose onto ceramide and generates glucosylceramide (GlcCer) on the cytosolic side of the Golgi. Within the Golgi lumen, B4GALT5 (*β*-1,4-galactosyltransferase 5) then transfers UDP-galactose onto GlcCer, producing lactosylceramide (LacCer), which is the shared precursor for most of the complex GSLs, such as globosides, lactosides, and gangliosides (Figure 3) [37]. SLC35A2 (solute carrier family 35 member A2) also plays a role in this pathway by transporting UDP-galactose from the cytosol into the Golgi lumen. In addition to de novo synthesis, ceramide can be generated through the salvage pathway (Figure 3) by recycling complex GSLs or re-acylating sphingosine [42,43].

Some bacterial toxins, such as Shiga toxins (Stxs), cholera toxin (Ctx), *Escherichia coli* heat-labile enterotoxin, tetanus neurotoxin (TeNT), and botulinum neurotoxins (BoNTs), are natural probes for GSLs since they specifically recognize GSLs as cellular receptors [44]. The recognition of GSLs by specific bacterial toxins offers a straightforward approach for conducting toxin-based genetic screens to comprehensively explore the biosynthesis pathway of GSLs. This strategy was challenging to achieve in the past due to limitations associated with traditional techniques. In this context, we focus on the recent screens involving two representative toxins, Stxs and Ctx, which recognize two distinct GSLs: globotriaosylceramide (Gb3, also known as CD77) and monosialotetrahexosylganglioside (GM1) as the receptor, respectively (Figure 3 and Table 1).

### 2.1. The Biology of Stxs

The Stx family includes the prototype Stx from *Shigella dysenteriae* and related Shiga-like toxins Stx1 and Stx2, produced by enterohemorrhagic *Escherichia coli* (EHEC) [45]. Stx1 differs from Stx by only one amino acid residue, whereas Stx2 represents distinct serotypes with ~56% sequence identity compared to Stx [45,46]. Belonging to the AB_5_ toxin superfamily, the Stx family is composed of an A chain (32 kDa) and five identical B chains (7.7 kDa each). The A chain is the enzymatic domain, which acts as an *N*-glycosidase that cleaves the host ribosomal RNA, while the five B chains form a pentameric ring and serve as the receptor binding domain. The A chain is connected to the B chain by inserting its C-terminus into the central pore of the B chain pentamer.

Once Stxs bind to the cellular receptor and enter cells through endocytosis by either clathrin-dependent or -independent pathways, they are sorted into the retrograde trafficking route and enter the *trans*-Golgi network (TGN). The A chain is processed by the host protease furin and cleaved into the enzymatic part A1 (27.5 kDa) and the B chain connecting part A2 (4.5 kDa). The A1 and A2 remain connected through an intramolecular disulfide bond between cysteines 242 and 261 residues. Stxs are further transported into the lumen of the ER, where the disulfide bond is reduced. The A1 part then crosses the ER membrane and enters the cytoplasm, utilizing the host ER-associated protein degradation (ERAD) machinery. The cytosolic Stxs eventually shut down protein synthesis by digesting ribosomal RNA and causing cell death [45,46].

The Stx B-chain pentamer specifically recognizes the carbohydrate moiety of Gb3 as its receptor [45,46,47,48]. The crystal structure suggests that each Stx B chain contains three Gb3 binding sites. Thus, one Stx holotoxin could maximally cluster fifteen Gb3 molecules on the cell surface [48]. Gb3 is the first member of the globo-series GSLs. The synthesis of Gb3 by transferring UDP-galactose onto LacCer is catalyzed by A4GALT (*α*-1,4-galactosyltransferase, also known as Gb3 synthetase, Figure 3) [49,50]. The expression of Gb3 in humans is highly restricted to the kidney, nervous system, microvascular endothelium, and a subset of germinal center B cells. In contrast, most other cell types do not express detectable levels of Gb3 [51,52,53,54]. The kidney-enriched Gb3 is responsible for the life-threatening post-diarrheal hemolytic uremic syndrome (D+HUS) induced by EHEC infection [46,55]. On the other hand, Gb3 lysosomal accumulation leads to a type of sphingolipidoses known as Fabry disease, which is the consequence of the loss-of-function of a lysosomal enzyme called α-galactosidase A, which is responsible for the degradation of Gb3 [56]. Enzyme replacement therapy is the only reliable treatment for Fabry disease nowadays [57,58]. In contrast, substrate reduction therapy is another promising approach involving the inhibition of Gb3 biosynthesis using small-molecule drugs such as ceramide analogs or imino-sugars, which holds promise as an alternative approach [38].

### 2.2. Genetic Screens for Stxs

In 2018, Tian et al. reported the first CRISPR-Cas9-mediated genome-wide screen for Stx1 and Stx2 [31]. The screen was conducted using the human bladder carcinoma 5637 cell line in a loss-of-function manner. The majority of the top-ranked hits overlapped between the Stx1 and Stx2 screens. Five top-ranked genes were key factors in the established Gb3 biosynthesis pathway: *SPTSSA*, *UGCG*, *B4GALT5*, *A4GALT*, and *SLC35A2*. Particularly, SPTSSA is a component of the serine palmitoyltransferase (SPT) complex on the ER membrane, which catalyzes the rate-limiting step in ceramide generation [37]. Other notable top-ranked hits shared by both screens include *UGP2* and *SPPL3*. UGP2 (UDP-glucose pyrophosphorylase 2) is the key enzyme that produces UDP-glucose, the substrate for GlcCer synthesis. SPPL3 (signal peptide peptidase-like 3) is a Golgi-localized protease implicated in the cleavage and activation of many glycosyltransferases [59,60]. Tian et al. then focused on investigating the other three newly identified factors, *LAPTM4A*, *TMEM165*, and *TM9SF2* [31].

*LAPTM4A* (lysosomal-associated protein transmembrane 4 A) was identified as a top hit in the screens (ranking No. 2 in the Stx1 screen and No. 1 in the Stx2 screen). However, its function had not been well characterized. Tian et al. found that knocking out *LAPTM4A* phenotypically mimicked knocking out *A4GALT* (Gb3 synthetase) in four aspects: (1) Both dramatically increased the cell resistance to Stx1 and Stx2, but not Ctx; (2) both abolished Stx cell surface binding but had no effect on Ctx binding; (3) both greatly reduced the expression level of Gb3, as measured by the mass spectrometry-based lipidomic assay; and (4) both induced the accumulation of the Gb3 precursor LacCer. However, the Golgi localization and the expression level of A4GALT were not altered in the *LAPTM4A*-knockout cells. Additionally, these properties of *LAPTM4A* are not shared with those of its homolog, *LAPTM4B* [31]. LAPTM4A is a small protein with 233 residues and four transmembrane domains, initially reported as an endosomal/lysosomal protein [61,62]. Tian et al. found that LAPTM4A is predominantly localized in the Golgi in multiple cell lines and physically interacts with A4GALT. Through the investigation of membrane topology and comparison of a panel of LAPTM4A/LAPTM4B chimeric proteins, the second lumenal domain of LAPTM4A was demonstrated to be critical for its function [31]. These results indicate that LAPTM4A is likely involved in the last step of Gb3 biosynthesis by serving as an essential co-factor for A4GALT’s enzymatic activity. However, the molecular basis of the interaction between LAPTM4A and A4GALT remains to be established.

*TMEM165* (transmembrane protein 165) encodes a multi-pass transmembrane protein that is Golgi-localized and has been proposed as a transporter for manganese ions (Mn^2+^) [63]. TMEM165 is critical for maintaining Mn^2+^ hemostasis, and its mutations have been linked to human disorders with defects in glycosylation [64,65], as Mn^2+^ is required for many Golgi-localized glycosyltransferases. Tian et al. found that *TMEM165*-deficient cells were more resistant and had lower cell surface binding to Stx and Ctx. In contrast to *LAPTM4A*, *TMEM165*-deficient cells had lower levels of Gb3 and Gb3 precursors and gangliosides, suggesting that TMEM165 affects the biosynthesis of GSLs globally. Consistent with previous reports, Tian et al. confirmed the Golgi localization of TMEM165. Furthermore, the downsides of *TMEM165* deficiency could be rescued by supplementing extra Mn^2+^, and *TMEM165*-deficient cells showed lower tolerance to Mn^2+^-induced cytotoxicity [31]. These findings experimentally confirmed the role of TMEM165 in regulating Mn^2+^ homeostasis.

*TM9SF2* (transmembrane 9 superfamily member 2) encodes a highly conserved but poorly characterized multi-pass transmembrane protein that has been reported to have endosomal or Golgi localization [66,67]. It has also been associated with multiple glycosylation pathways, including heparan sulfate proteoglycan biosynthesis [67]. Tian et al. verified the Golgi-localization of TM9SF2 in multiple cell lines and found that *TM9SF2*-knockout cells had a lower level of surface heparan sulfate. Similar to *TMEM165*, *TM9SF2*-knockout cells expressed a lower level of GSLs, indicating that loss of *TM9SF2* causes a global disruption in GSL biosynthesis and contributes to the resistance to both Stx and Ctx [31]. The detailed mechanism of how *TM9SF2* is involved in glycosylation, whether similar to *TMEM165* [54], remains to be established.

In 2019, Yamaji et al. reported an independent CRISPR screen for Stx1 in HeLa cells [68]. The screen once again identified well-established genes involved in GSL biosynthesis and the novel factors *LAPTM4A*, *TMEM165*, and *TM9SF2*. Using radioisotope labeling and thin-layer chromatography, Yamaji et al. demonstrated the requirement for *LAPTM4A* but not *LAPTM4B* in the last step of Gb3 synthesis. They further provided direct evidence showing that the enzymatic activity of A4GALT in cell lysates was greatly reduced when *LAPTM4A* was absent [68]. Yamaji et al. also showed that *TM9SF2* is involved in Gb3 biosynthesis (likely through A4GALT) by its conserved C-terminus across all transmembrane 9 superfamily members (TM9SF1, TM9SF2, TM9SF3, and TM9SF4) [68]. In 2021, the same group reported a related screen on Vero cells derived from green monkeys, and the knockout was generated by a library targeting the human genome [69]. Although the degree of gene enrichment was less than the compatible screen in human cells, this screen still identified major players in the GSLs pathway, including *LAPTM4A* and *TM9SF2* [69]. In 2020, Majumder et al. performed a similar screen on HeLa cells [70]. In addition to the same set of factors (including *LAPTM4A* and *TM9SF2*), a transcription factor, *AHR* (aryl hydrocarbon receptor), was uniquely identified, which may regulate Gb3 biosynthesis by regulating the expression of several known factors such as *SPTSSA* [70].

In 2018, Pacheco et al. conducted a unique screen directly using EHEC. This carries an additional virulence factor besides Stx and is known as the type III secretion system (T3SS) in the human intestinal epithelial HT29 cell line. HT29 cells were chosen because they are sensitive to EHEC co-culture but resistant to purified Stx [71]. Interestingly, many of the identified genes in this screen were involved in Gb3 biosynthesis (including *LAPTM4A* and *TM9SF2*), suggesting the potential role of Gb3 in the virulence of both Stx and T3SS. This finding further emphasizes that targeting factors in the GSL biosynthesis pathway is promising for designing new drugs to combat Stx and EHEC infections [71].

Another unique screen was reported by Kono et al., which specially focused on the ceramide salvage pathway [72]. The screen was carried out in HeLa cells with a de novo ceramide synthesis defect by knocking out the key gene *SPTLC1* (serine palmitoyltransferase long chain base subunit 1). The *SPTLC1*-knockout cells are unable to generate 3-Keto-dihydrosphingosine from serine and palmitoyl-CoA. Then, sphingosine-1-phosphate (S1P) was added to the culture medium to activate the salvage pathway and restore the expression of Gb3. The CRISPR screen using Stx2 as killing stress under this condition successfully identified several important genes. These include ceramide synthase *CERS2*, well-known genes in the Gb3 biosynthesis pathway (*UGCG*, *B4GALT5*, *A4GALT*, and *SLC35A2*), and three newly identified factors (*LAPTM4A*, *TMEM165*, and *TM9SF2*). Additionally, two phosphatases, *PLPP3* (phospholipid phosphatase 3, also known as *PPAP2B*) and *SGPP1* (S1P phosphatase 1), were also identified through this screening process [72]. Kono et al. found that the cell surface-expressed PLPP3 is important for the uptake of extracellular S1P by dephosphorylating S1P into sphingosine. Then, the cellular sphingosine is rephosphorylated to S1P and further dephosphorylated by SGPP1 for ceramide synthesis (Figure 3) [72].

### 2.3. Ctx and the Related Screens

Ctx is the major virulence factor produced by toxigenic strains of *Vibrio cholerae* [73]. Similar to Stxs, Ctx also belongs to the AB_5_ toxin superfamily and shows a similar overall architecture. Upon binding to the cell surface and endocytosis, Ctx undergoes retrograde trafficking and releases its enzymatic A chain across the ER membrane [74]. The cytosol-exposed A chain deactivates the GTP hydrolase activity of the G_S_ alpha subunit through an ADP-ribosylation reaction and causes the continuous expression of 3′,5′-cyclic AMP (cAMP). This, in turn, triggers a series of consequences and eventually leads to the opening of the cAMP-dependent chloride channel CFTR (cystic fibrosis transmembrane conductance regulator) [75,76,77]. This process is responsible for the pathogenic effects caused by cholera infection, such as rapid fluid loss and rice-water stool [78].

The Ctx B-chain pentamer specifically recognizes the carbohydrate moiety of GM1, particularly GM1a in the a-series of gangliosides, as its receptor (Figure 3) [79]. Structural studies suggest that one Ctx holotoxin could maximally cluster five GM1a molecules on the cell surface [80]. Thus, the Ctx B-chain pentamer has been widely used as a probe for studying ganglioside biology [81]. The biosynthesis of GM1a from LacCer requires three steps (Figure 3): (1) adding CMP-sialic acid to LacCer and generating GM3 by ST3GAL5 (ST3 *β*-galactoside *α*-2,3-sialyltransferase 5); (2) adding UDP-*N*-acetylgalactosamine (UDP-GalNAc) to GM3 and generating GM2 by B4GALNT1 (*β*-1,4-*N*-acetylgalactosaminyltransferase 1); and (3) adding UDP-galactose to GM2 and generating GM1a by B3GALT4 (*β*-1,3-galactosyltransferase 4) [82]. Unlike Gb3, GM1 is widely expressed in human tissues and is involved in multiple essential functions [82].

In 2011, Guimaraes et al. reported a genetic screen for Ctx using a retroviral insertion-based loss-of-function strategy in human haploid KBM7 cells [83]. Since Ctx itself cannot sufficiently kill cells, Guimaraes et al. engineered a lethal chimera toxin by fusing the enzymatic domain of diphtheria toxin (DTA) to the Ctx A chain using the sortase ligation method. DTA is also an ADP-ribosylation enzyme, but it specifically modifies a unique residue called diphthamide in eukaryotic elongation factor 2 (eEF-2), thereby blocking protein synthesis and inducing cell death [84,85]. The screen identified the genes involved in diphthamide biosynthesis that are responsible for DTA, as well as a series of genes involved in GM1a biosynthesis (e.g., *UGCG*, *SLC35A2*, *B3GALT4*, and *ST3GAL5*). ST3GAL5 is the GM3 synthase, and GM3 is the shared precursor for gangliosides from the a-, b-, and c-series but not from the 0-series (e.g., GM1b, Figure 3) [86]. Guimaraes et al. found that the *ST3GAL5*-knockout cells were resistant to Ctx. However, Ctx could still bind to a subset of cells (5–10%) at levels comparable to wild-type cells. They speculated that in the absence of ST3GAL5, an alternative ganglioside synthesis pathway for the 0-series can be initiated in a cell cycle-dependent manner. In this case, GM1b acts as an alternative Ctx receptor [83].

In 2014, Gilbert et al. reported another set of genetic screens for the Ctx-DTA chimera toxin using the emerging CRISPRi and CRISPRa approaches [26]. The results of the CRISPRi-based loss-of-function screen suggested that downregulating the genes involved in GM1a biosynthesis (e.g., *B3GALT4* and *ST3GAL5*) leads to the protection of Ctx-DTA. In contrast, downregulating the genes involved in the biosynthesis of other gangliosides, including GM1b (e.g., *ST3GAL2*), resulted in sensitization to Ctx-DTA. The CRISPRa-based gain-of-function screen yielded results consistent with the CRISPRi screen. In addition, upregulating the genes involved in the biosynthesis of lactosides/neolactosides (e.g., *B3GNT5*) had protective effects. Both CRISPRi and CRISPRa screens suggested that protection against Ctx-DTA was apparently caused by diverting the shared precursor LacCer away from GM1a synthesis to other branches (Figure 3) [26]. These recent genetic screens expand our understanding of the complex and branched GSL biosynthesis pathways.

## 3. Factors Involved in Protein Glycosylation

Protein glycosylation is a common form of modification that can occur both co-translationally and post-translationally [87]. There are several types of glycosylation based on the chemistry of which atom from an amino acid residue is attached to the carbohydrate moieties. These types include *N*-linked (where the nitrogen atom provided by asparagine or arginine is attached), *O*-linked (where the oxygen atom provided by serine or threonine is attached), *C*-linked (where the carbon atom provided by tryptophan is attached), and a special type called glypiation (where the C-terminus of the protein is attached by a glycosylphosphatidylinositol (GPI) anchor) (Figure 4).

Glycosylation influences the folding and stability of glycoproteins and is critical for maintaining many cellular functions, such as cell-to-cell communication and adhesion [88]. Congenital disorders of glycosylation (CDG) are a family of human diseases characterized by genetic defects in the glycosylation process, and effective therapies for these disorders are currently lacking [89]. The core reaction of protein glycosylation is initiated in the ER, and the complex structures are synthesized in the Golgi. This enzymatic process requires a large number of enzymes, such as glycosyltransferases, glycosidases, and transporters [90]. Many CDGs have been associated with the loss-of-function mutations identified in these enzymes and the key regulatory factors that control specificity and/or activity [91]. Therefore, a deep understanding of the fundamental glycosylation pathways at the molecular level can help pinpoint therapeutic approaches.

Lectins are a group of proteins that exhibit high specificity in binding to carbohydrate moieties [92]. Many toxins exhibit lectin activity and utilize glycoproteins as their receptors. Examples of such toxins include *Clostridioides difficile* toxins A (TcdA) and B (TcdB), *Escherichia coli* subtilase cytotoxin (SubAB), *Staphylococcus aureus* bi-component pore-forming toxins, insecticidal toxin complexes (Tc toxins), cholesterol-dependent cytolysins (CDCs), pertussis toxin (Ptx), and the plant toxin ricin. These toxins can serve as powerful probes to comprehensively survey protein glycosylation pathways using genome-wide genetic screens. In this section, we focus on recent toxin-based screens for four major glycosylation pathways: *N*-linked glycosylation, *O*-linked glycosylation, biosynthesis of sulfated glycosaminoglycans (sGAGs, a special type of *O*-linked glycosylation), and biosynthesis of GPI anchor (Figure 4 and Table 1).

### 3.1. Ricin and Related Screens for N-Linked Glycosylation

Ricin, a potential bioterrorism agent, is produced from the seeds of the castor oil plant (*Ricinus communis*) and has been classified as Biological Select Agents or Toxins (BSATs) by the United States Department of Health and Human Services (HHS) [93]. Ricin is composed of an A chain (32 kDa) and a B chain (34 kDa) connected by an intermolecular disulfide bond [94]. Despite being evolutionarily distinct from Stxs, ricin and Stxs share some common features: (1) Their A chains both function as *N*-glycosidase and cleave the host ribosomal RNA at the same site, and (2) they both undergo retrograde trafficking pathways and enter the cytoplasm through the ER membrane. The major difference in the mechanism of action between ricin and Stxs is the receptor recognition executed by their respective B chains. Ricin’s B chain is a lectin that broadly binds to the carbohydrate moieties of *N*-linked glycoproteins containing terminal galactose, or GalNAc [95,96].

Ricin has been screened on eukaryotic cells multiple times using various approaches [26,31,96,97,98,99,100,101]. In 2018, alongside the Stxs screens, Tian et al. also carried out a genome-wide CRISPR-Cas9-mediated screen for ricin in parallel [31]. This screen was conducted in HeLa cells and identified major factors involved in *N*-linked glycosylation. These factors include the ER-localized glycosyltransferases for producing the dolichol-linked precursor oligosaccharide (e.g., *ALG5*, *ALG6*, *ALG8*, and *MOGS*), the ER-localized transferase catalyzing the transfer of the precursor oligosaccharide to a protein (e.g., *OST4*), and the Golgi-localized glycosidases and glycosyltransferases converting the high-mannose precursor oligosaccharide to *N*-acetylglucosamine (GlcNAc)-containing complex-type glycans (e.g., *MAN1A2*, *MAN2A1*, *MGAT1*, and *MGAT2*, Figure 4) [31]. The screen also identified key factors in the fucosylation pathway, including cytosolic enzymes for the biosynthesis of GDP-fucose (e.g., *TSTA3* and *GMDS*), and the Golgi-localized GDP-fucose transporter *SLC35C1* and fucosyltransferase *FUT4* [31]. These results are consistent with the established view that ricin mainly recognizes terminal galactoses on the *N*-linked glycans as primary receptors. Fucosylation plays a critical role, possibly because transferring fucose to GlcNAc prevents the sialyation of terminal galactoses [96,97,100,101,102].

Tian et al. also identified *TMEM165* and *TM9SF2* as shared factors between Stxs and ricin, while *LAPTM4A* is specific to Stxs [31]. They found that *TMEM165*- and *TM9SF2*-deficient cells are more resistant to ricin during short-term exposure (20–30 h), but this resistance diminished when the exposure time was extended to 40–50 h. This result suggests that the loss of *TMEM165* or *TM9SF2* causes a mild reduction in sensitivity to ricin, which is consistent with their indirect roles in regulating *N*-linked glycosylation.

### 3.2. TcdA and Related Screens for Biosynthesis of sGAGs

*Clostridioides difficile* produces two major virulence factors: TcdA and TcdB [103]. Both toxins belong to the large clostridial toxin (LCT) family and consist of four functional domains: (1) an N-terminal enzymatic domain with glucosyltransferase activity; (2) an auto-activation domain; (3) a central domain with both receptor binding and transmembrane delivery activities; and (4) a C-terminal domain with lectin activity and known as combined repetitive oligopeptides (CROPs) [3,104,105,106,107,108,109]. The CROPs domain of TcdA has been shown to broadly interact with the cell surface carbohydrate moieties containing the galactose-*β*-1,4-GlcNAc motif [110,111,112]. Although the CROPs domains may initiate the attachment of LCT family toxins to the surface of host cells, CROPs-independent receptors have also been identified [32,35,113,114,115].

In 2019, Tao et al. reported a genome-wide CRISPR-Cas9-mediated screen using a truncated form of TcdA without the CROPs domain [33]. The screen identified the low-density lipoprotein receptor (LDLR) as a host factor that mediates TcdA cellular entry through rapid endocytosis and recycling. Surprisingly, the screen also revealed factors in the biosynthesis pathway of sGAG-linked proteoglycan [116]. These factors include the Golgi-localized glycosyltransferases (e.g., *XYLT1*, *XYLT2*, *B4GALT7*, *B3GALT6*, *EXT1*, *EXT2*, *EXTL1*, *EXTL2*, and *EXTL3*), the *N*-sulfotransferase (e.g., *NDST1* and *NDST2*), the *O*-sulfotransferase (e.g., *HS2ST*, *HS3ST*, and *HS6ST1*), and the transporter for the activated form of sulfate *SLC35B2* (Figure 4) [33]. These results suggest that the sulfation groups in sGAGs may contribute to TcdA binding to cells. Furthermore, the screen identified *TMEM165* again, indicating its additional functions in sGAG biosynthesis [33]. Notably, in 2021, another related screen reported by Zhou et al. used a similar strategy for *Clostridium novyi* alpha-toxin (TcnA, also belonging to the LCT family). They obtained a similar set of sGAG-related genes [117].

In addition to forming and organizing the extracellular matrix, sGAGs exhibit a variety of essential functions, such as signaling transduction and organ development [118]. Defects in sGAG factors have been linked to a large panel of CDGs that occur in multiple tissues, such as Ehlers-Danlos syndrome, Baratela-Scott syndrome, and Langer-Giedion syndrome [119]. These recent toxin-based genetic screens have fully revealed key players in the sGAG biosynthesis pathway, providing not only new biological insights but also potential therapeutic strategies for addressing these disorders.

### 3.3. TcdB and Related Screens for Biosynthesis of GPI Anchor

TcdB has been shown to recognize two independent receptors, chondroitin sulfate proteoglycan 4 (CSPG4) and frizzled receptors (FZDs), through distinct binding interfaces [113,120,121,122,123,124,125]. Taking advantage of recent sequencing and functional analyses, TcdB subtypes with 3–15% amino acid sequence variations and receptor-binding divergence from the archetype (designated TcdB1) have been identified [126,127]. Intragenic micro-recombination occurring around the receptor-binding regions has been proposed as the driving force for the rapid evolution and diversification of TcdB subtypes [35,127].

In 2022, two back-to-back papers independently reported genome-wide CRISPR-Cas9-mediated screens using subtype TcdB4, and both screens identified tissue factor pathway inhibitor (TFPI) as a novel TcdB receptor [35,114]. TFPI has two major splicing forms: the soluble form TFPIα and the GPI-anchored form TFPIβ [128,129]. TFPIβ is the major form that contributes to TcdB4 cellular binding and entry. Consistently, several well-known enzymes involved in the GPI pathway [130] were also enriched in the screens, such as glycosyltransferases (e.g., *PIGA*, *PIGB*, *PIGC*, *PIGM*, *PIGP*, *PIGQ*, *PIGV*, *PIGX*, *PIGY*, and *DPM1*), ethanolaminephosphate transferase *PIGF*, inositol acyltransferase *PIGW*, and GPI transamidase components (e.g., *PIGS*, *PIGU*, and *GPAA1*) (Figure 4).

In the human proteome, more than 150 proteins use the GPI anchor strategy to display on the cell surface [130]. Defects in GPI biosynthesis have been linked to rare diseases such as hyperphosphatasia with mental retardation syndrome (HPMRS) and paroxysmal nocturnal hemoglobinuria (PNH) [131]. These recent toxin-based genetic screens identified some genes with unknown functions, such as *ZNF619*, *RNF41*, *PNMA1*, *C19orf67*, and *C14orf132* [35,114]. However, whether these genes contribute to GPI biosynthesis remains to be established.

### 3.4. Other Toxin-Based Screens for Glycosylation Pathways

In 2019, Yamaji et al. reported a CRISPR screen for the *Escherichia coli* toxin SubAB [132], which is known to utilize terminal sialic acid on glycoproteins as a receptor [133]. Consistently, the screen identified genes involved in both *N*-linked glycosylation (e.g., *MGAT1* and *MAN2A1*) and *O*-linked glycosylation (e.g., *C1GALT1* and *C1GALT1C1*), as well as *TMEM165* and a predicted Golgi-localized zinc transporter, *SLC39A9*. Yamaji et al. found that the loss of *SLC39A9* reduced both complex-type *N*-linked glycans, likely through regulating glycosidase MAN2A1, and core 1 *O*-linked glycans by reducing the key glycosyltransferase C1GALT1. Furthermore, when a key residue (H155) for the predicted Zn^2+^ transporter activity had been mutated, SLC39A9 lost its ability to regulate glycosylation [132]. Future investigations are required to understand the mechanisms by which SLC39A9 regulates Zn^2+^ hemostasis and why Zn^2+^ is involved in glycosylation.

In 2020, Tromp et al. reported two CRISPR screens for the *Staphylococcus aureus*-produced bi-component pore-forming toxins Panton-Valentine leukocidin (PVL) and γ-haemolysin CB (HlgCB) [134]. These are known to utilize C5AR1 and C5AR2, two G-protein coupled receptors (GPCRs) with post-translational modifications, as receptors [135,136]. The screens identified the known toxins receptor *C5AR1* as a positive control and the genes involved in the tyrosine sulfation pathway (e.g., *SLC35B2*, *PAPSS1*, and *TPST2*) as shared host factors between PVL and HlgCB, while the genes involved in the sialyation pathway (e.g., *SLC35A1* and *CMAS*) were unique host factors for HlgCB but not PVL. Tromp et al. further found that sialyation-deficient cells had a lower surface expression level of CXCR2 [134], another GPCR recognized as a receptor for the other two leukotoxins LukED and HlgAB [136,137]. The detailed mechanism by which sialyation regulates surface protein expression remains to be investigated.

In 2020, Drabavicius et al. reported a CRISPR screen in near-haploid HAP1 cells for *Streptococcus intermedius*-produced CDC intermedilysin (ILY) [138], which has been known to utilize a GPI-anchored protein CD59 as a receptor [139,140]. This screen, along with a similar one in HeLa cells reported by Shahi et al. [141], successfully identified the known ILY receptor *CD59* and a whole bunch of host factors in the GPI biosynthesis pathway (e.g., *PIGA* and *PIGB*). Drabavicius et al. also identified a panel of genes involved in various cellular processes. These include genes related to *N*-linked glycosylation (e.g., *MGAT1* and *MOGS*), GSLs biosynthesis (e.g., *UGCG* and *B4GALT5*), sGAGs biosynthesis (e.g., *EXT2* and *B3GALT6*), nucleotide sugar metabolism (e.g., *GALE*, *UGP2*, and *UXS1*), as well as *TM9SF2* and a subset of genes with unclear functions in ILY intoxication (e.g., *C12orf43*, *C12orf49*, *TMEM30A*, and *PDCD10*). This screen was so vigorous that it revealed a whole bunch of established factors in multiple glycosylation pathways, possibly due to the enhanced response of haploid HAP1 cells to CRISPR-mediated loss-of-function mutagenesis. Therefore, it is worth paying extra attention to the newly identified factors and their potential roles in glycosylation.

The insecticidal toxin complexes (Tc toxins) produced by entomopathogenic bacteria also exhibit lectin activities [142,143]. Several genetic screens have been conducted to investigate Tc toxins from multiple species using various platforms [144,145]. Through these screens, several genes involved in *N*-linked glycosylation (e.g., *MGAT1* and *MAN1A1*), GPI biosynthesis (e.g., *MPDU1*), and sGAG biosynthesis (e.g., *EXTL3*, *SLC35B2*, *B3GALT6*, and Drosophila *sgl*) have been identified. Notably, *TMEM165* and *TM9SF2* were identified in a screen for a Tc toxin interacting with sGAGs [144], further confirming their broad involvement in glycosylation.

## 4. Factors Involved in Membrane Vesicle Trafficking

Many toxins, such as the previously mentioned Stxs, Ctx, ricin, SubAB, pertussis toxin, as well as *Pseudomonas aeruginosa* exotoxin A (EtA), *Salmonella Typhi* typhoid toxin, and cytolethal distending toxin (Cdt), exploit retrograde trafficking routes to enter the cytosol. Consistently knocking out or knocking down key factors that regulate trafficking pathways, as well as treatment with small-molecule inhibitors that disrupt the function of the Golgi apparatus (e.g., Brefeldin A) or specifically block trafficking pathways (e.g., Retro-1 and Retro-2 [146]) reduce toxins’ induced toxicities. Therefore, these toxins can be used as probes (different toxins could hijack redundant pathways [99]) to study the membrane vesicle trafficking pathways (Table 1), which are essential for cellular functions, and the defects in these pathways are related to various human diseases [147].

### 4.1. Screens Using Ricin as a Probe

In 2013, Bassik et al. reported a genome-wide screen for ricin using a short hairpin RNA (shRNA)-based RNAi approach [98]. The screen was primarily conducted using a single-shRNA library (25 shRNAs per gene) to identify the highly confident genes and shRNAs. This was followed by a double-shRNA library based on the results from the primary screen (knockdown two genes simultaneously) to systematically measure the genetic interactions. Bassik et al. identified a set of top genes, including both protective hits (knockdown causing ricin resistance) and sensitizing hits (knockdown causing ricin sensitization) [98]. These top genes were enriched in membrane vesicle trafficking pathways, such as the components of COPII (coat protein complex II, which facilitates ER-Golgi anterograde transport, e.g., *SEC24A* and *SEC24B*), TRAPP (transport protein particle, which facilitates ER-Golgi anterograde transport, e.g., *TRAPPC8* and *TRAPPC11*), and GARP (Golgi-associated retrograde protein, which facilitates endosome-Golgi retrograde transport, e.g., *VPS53* and *VPS54*) as protective genes. In contrast, the components of COPI (coat protein complex I, which facilitates endosome-Golgi-ER retrograde transport, e.g., *ARCN1* and *COPZ1*) and Retromer (which facilitates endosome-Golgi retrograde transport, e.g., *VPS35* and *VPS26A*) were identified as sensitizing genes. The finding that losing COPII and TRAPP components causes strong protection against ricin, which was unexpected. This suggested that shutting down ER-budding and anterograde trafficking may cause Golgi dysfunction. By analyzing genetic interactions through the secondary screen, Bassik et al. identified previously poorly characterized genes *C4orf41*, *KIAA1012*, and *C5orf44* that function as TRAPP interactors, thereby defining two types of TRAPP complexes with distinct compositions and opposite functions in trafficking pathways [98].

A CRISPR-based genetic screen for ricin reported by Tian et al. also revealed several components of the GARP complex (e.g., *VPS51*, *VPS52*, *VPS53*, and *VPS54*), genes involved in Golgi-ER trafficking (e.g., *GOSR1*, *NAPG*, *NBAS*, *STX5*, and *ARL5B*), and the ERAD factors that facilitate toxin crossing the ER membrane (e.g., *UBE2G2*). Interestingly, in addition to its association with glycosylation pathways, *TM9SF2*, the shared host factor between ricin and Stxs, was demonstrated to be involved in intracellular vesicular transport. Tian et al. found that *TM9SF2*-knockout cells exhibited large vacuoles that colocalized with Rab7 within the cytosol and showed abnormal endosomal trafficking across distinct cell types by monitoring the exogenously loaded fluorescently labeled lipids [31]. Consistently, Yamaji et al. showed the disruption of the TGN in *TM9SF2*-knockout cells and further proposed that TM9SF2 may be required for the retrograde trafficking of glycosyltransferases such as A4GALT [68]. The mechanism underlying the TM9SF2-trafficking axis remains to be established.

### 4.2. Screens Using Stx as a Probe

In 2017, Selyunin et al. reported a screen for Stxs utilizing a small interfering RNA (siRNA)-based RNAi approach on A4GALT-overexpressing HeLa cells [148]. The screen identified the Golgi-localized protein UNC50 as required for Stx2 endosome-Golgi retrograde trafficking, likely by recruiting the trafficking factor GBF1 (Golgi Brefeldin A resistant guanine nucleotide exchange factor 1) to the Golgi [148]. In the CRISPR screens reported by Sakuma et al., in addition to *UNC50*, other trafficking factors were identified (e.g., *SYS1* and the components of the GARP complex) [68,69]. Particularly, *SYS1* (SYS1 Golgi trafficking protein) was previously identified as a host factor for *Staphylococcus aureus* α-hemolysin (αHL) by regulating the expression of the toxin receptor ADAM10 (ADAM metalloproteinase domain 10) [149]. When *SYS1* was knocked out in Vero cells, the TGN showed abnormal morphology and function, resulting in glycosylation defects that made the cells more resistant to Stx [69]. Although UNC50 and SYS1 share some similar features, overexpressing *UNC50* in the *SYS1* knockout cells failed to compensate for the defects in glycosylation, suggesting that these two factors likely have different functions [69].

In the screens for Stxs reported by Tian et al., the trafficking factor *ARCN1* (a COPI component) was identified as a top-ranked hit, whereas neither *UNC50* nor *SYS1* was identified [31]. This screen recovered fewer trafficking factors, possibly due to the screening strategy relying on long-term toxin treatment (72 h incubation) so that the minor reductions caused by defects in trafficking could be masked. In 2020, Kouzel et al. compared the RNA sequencing profiles between ACHN and Caki-2 cells—two human kidney cell lines that both have Gb3 expression but show different Stx sensitivities [150]. Upon the Stx2 challenge, many trafficking factors showed differential expression, including *RAB5A*, *TRAPPC6B*, and *YKT6*. Notably, neither *UNC50* nor *SYS1* was identified as differentially expressed genes [150]. The dependence of different trafficking factors on different toxins among various cells or tissues remains to be clarified.

### 4.3. Screens Using EtA as a Probe

EtA is another toxin that undergoes the retrograde trafficking pathway. Its enzymatic domain acts as an ADP-ribosylation enzyme to deactivate eEF-2, similar to DTA [151]. In 2014, Tafesse et al. reported a genetic screen for EtA using a retrovirus-based mutagenesis strategy in haploid KBM7 cells [152]. The screen successfully identified genes involved in diphthamide biosynthesis (e.g., *DPH1*, *DPH2*, and *DPH4*) corresponding to EtA’s enzymatic activity. A panel of trafficking factors (e.g., components of the GARP complex and the Golgi-ER trafficking factor *KDELR1*) were also identified. The screen also revealed a new factor, *GPR107*, a poorly characterized GPCR that had been identified but not characterized in another genetic screen for ricin [101]. Tafesse et al. found that GPR107 is a TGN-localized trafficking factor, and its N-terminal domain, which needs to be activated by furin protease, is critical for retrograde transport [152].

In 2011, Moreau et al. reported two RNAi-based genetic screens for EtA and ricin, respectively [99]. The screens identified a large set of genes involved in membrane vesicle trafficking. This included 65 ricin-specific factors (e.g., components of TRAPP complex), 69 EtA-specific factors (e.g., components of the Retromer and *KDELR*), and 44 shared factors between two toxins (e.g., components of the GARP complex and *STX16*). These results demonstrated the genetic complexity of the retrograde trafficking pathway, which is likely not only hijacked by different toxins but also underlies the complexity of cellular membrane-bound compartments [99].

### 4.4. Screens Using Other Toxins as Probes

In the SubAB screen reported by Yamaji et al., in addition to the genes involved in glycosylation, many trafficking factors were identified (e.g., components of the GARP complex, COG complex, and *UNC50*) [132]. The screen also revealed *KDELR1*, *KDELR2* (KDEL receptors 1 and 2), and *JTB* (jumping translocation breakpoint, which were also enriched in a ricin screen [31]). Knocking out these factors did not affect SubAB binding to the cell surface but did suppress the cleavage of the toxin’s intracellular substrate, suggesting their involvement in toxin trafficking [132].

In 2019, Chang et al. reported a CRISPR screen using typhoid toxin as a probe [153]. The screen identified known toxin trafficking factors (e.g., components of the GARP complex, COG (conserved oligomeric Golgi) complex, COPI complex, *UNC50*, and *GPR107*) and the ERAD factors (e.g., *SEL1L* and *SYVN1*). The screen also revealed *TMED2* (transmembrane p24 trafficking protein 2) as a unique Golgi-ER trafficking factor for typhoid toxin, possibly acting as a specific cargo receptor and working together with the COPI complex [153].

## 5. Factors Involved in Unique Pathways

In 2009, Carette et al. developed a retrovirus-based genetic screen platform that relies on haploid KBM7 cells [18]. This platform has been applied to screen the host factors for diphtheria toxin (DT) and anthrax-diphtheria chimera toxin (anthrax protective antigen (PA) plus anthrax lethal factor N-terminal (LFN) fused with DTA (PA-LFN-DTA)). The screens identified the known toxin receptors (*ANTXR2* for PA and *HBEGF* for DT, respectively) and genes involved in diphthamide biosynthesis (e.g., *DPH1*, *DPH2*, and *DPH5*) corresponding to the enzymatic activity of DTA. A previously uncharacterized gene, *WDR85*, which is the ortholog of the yeast gene *YBR246W*, was also identified. Carette et al. found that *WDR85* was required for the toxicities but not for the entry of DT, PA-LFN-DTA, or EtA. Furthermore, *WDR85* was demonstrated to be involved in diphthamide biosynthesis [18] and has since been renamed *DPH7*.

By analyzing the genetic interactions through the RNAi-based screen for ricin [98], Bassik et al. found that knocking out the ribosomal protein *RPS25* uniquely led to ricin resistance, and *RPS25* formed a genetic cluster with the transcription factors *ILF2* and *ILF3*. The strong buffering interactions between *RPS25* and *ILF2/3* suggest that they may physically interact together to control the translation of certain factors in ricin intoxication. Bassik et al. also identified two previously uncharacterized genes, *WDR11* and *C17orf75*, which form a genetic cluster and interact physically. The WDR11-C17orf75 protein complex may regulate ricin degradation through the autophagy pathway [98].

In 2016, Tao et al. reported a CRISPR screen for TcdB, and the members of the ER membrane protein complex (EMC), such as *EMC1*, *EMC3*, *EMC4*, *EMC5*, and *EMC6*, were identified as TcdB host factors. Loss of *EMC* reduces the expression level of the toxin receptor FZDs [113]. To figure out the full scope of EMC-dependent proteins besides FZDs, Tian et al. carried out a follow-up study using unbiased quantitative proteomic analysis coupled with tandem mass tag labeling and mass spectrometry [154]. Subsets of EMC-dependent and EMC-independent membrane proteins were identified by comparing the membrane protein profiles in wild-type and EMC-knockout cells. Bioinformatic analysis revealed a common feature of EMC-dependent proteins: their transmembrane domains contain polar/charged residues. Introducing or deleting these polar/charged residues can switch the EMC dependency [154].

Recently, Anwar et al. conducted an image-based siRNA screen for anthrax toxin [155]. The screen focused on cell-surface proteins and trafficking factors, utilizing the cleavage of a toxin substrate as a readout. As a result, Anwar et al. identified another TMED member, *TMED10*, which is essential for anthrax toxin oligomerization on the cell surface. It has been established that TMED10 forms a heterodimer with TMED2. In addition to its role as a cargo receptor in Golgi-ER trafficking, the TMED2/10 complex was demonstrated to facilitate non-vesicular lipid transfer at the ER-Golgi membrane contact site, thereby controlling the formation of plasma membrane lipid nanodomains [155].

Another recently reported CRISPR screen for PVL by Jeon et al. was conducted in differentiated macrophages [156]. Besides the known PVL receptor *C5AR1*, *FBXO11* (F-box protein 11) was also enriched in the screen. Jeon et al. found that *FBXO11*-knockout cells exhibited reduced transcription of *C5AR1* that could be restored by lipopolysaccharide (LPS) priming. FBXO11 was further demonstrated to regulate the inflammasome pathway and the expression of IL-1β through BCL6-dependent and BCL6-independent transcriptional regulations [156].

## 6. Summary

The toxin-based genetic screens discussed above have been summarized in Table 1. It highlights their screen strategies (e.g., mutagenesis approaches, screen scale, screen formats, and cell models), the identified host factors with known functions, and the newly identified host factors with new functions in fundamental biological pathways. Notably, there is another long list of wonderful studies on toxin–host interactions that have successfully identified toxin–host factors, but they are not discussed here for the following reasons: (1) the identified factors directly interact with toxins, such as receptors; (2) the factors were identified but not functionally characterized; and (3) the factors or pathways were identified by other approaches, such as affinity purification (e.g., pull-down), active labeling (e.g., chemical crosslinking and enzymatic labeling), multi-omics analyses (e.g., proteomics, transcriptomics, lipidomics, and glycomics), and so on.

**Table 1 bioengineering-10-00884-t001:** Toxin-based genetic screens are discussed in this review.

Screens	Toxins	Screen Strategy *	Identified Host Factors and Functions
Reported by Tian et al. [31]	Stx1 Stx2 ricin	CRISPR-Cas9 Genome-wide Loss-of-function Human 5637 and HeLa cells	Known genes involved in Gb3 biosynthesis. Known genes involved in N-linked glycosylation and fucosylation pathways. Known genes involved in membrane vesicle trafficking. LAPTM4A: interacts with A4GALT in the Golgi and is required for A4GALT’s enzymatic activity. TMEM165: acts as a Golgi Mn2+ transporter and globally regulates glycosylation. TM9SF2: globally involves in the Golgi glycosylation and membrane vesicle trafficking.
Reported by Yamaji et al. [68]	Stx1	CRISPR-Cas9 Genome-wide Loss-of-function Human HeLa cells	Known genes involved in Gb3 biosynthesis. Known genes involved in membrane vesicle trafficking. LAPTM4A and TMEM165. TM9SF2: involves in Gb3 biosynthesis likely through A4GALT; and involves in membrane vesicle trafficking.
Reported by Sakuma et al. [69]	Stx1	CRISPR-Cas9 Genome-wide Loss-of-function Green monkey Vero cells **	Known genes involved in Gb3 biosynthesis. Known genes involved in membrane vesicle trafficking. LAPTM4A and TM9SF2. SYS1: is required for maintaining Golgi morphology and function.
Reported by Majumder et al. [70]	Stx	CRISPR-Cas9 Genome-wide Loss-of-function SPTLC1-knockout Human HeLa cells	Known genes involved in Gb3 biosynthesis. Known genes involved in membrane vesicle trafficking. LAPTM4A, TMEM165, and TM9SF2. AHR: regulates the expression of Gb3 synthesis factors.
Reported by Pacheco et al. [71]	EHEC ***	CRISPR-Cas9 Genome-wide Loss-of-function Human HT29 cells	Known genes involved in Gb3 biosynthesis. LAPTM4A and TM9SF2.
Reported by Kono et al. [72]	Stx	CRISPR-Cas9 Genome-wide Loss-of-function SPTLC1-knockout HeLa cells supplementation with S1P	Known genes involved in Gb3 biosynthesis. LAPTM4A, TMEM165, and TM9SF2. PLPP3: regulates the uptake of extracellular S1P. SGPP1: dephosphorylates S1P for ceramide synthesis.
Reported by Selyunin et al. [148]	Stx1 Stx2	RNAi (siRNA) Genome-wide Loss-of-function A4GALT-overexpressing HeLa cells	UNC50: is required for Stx2 endosome-Golgi retrograde trafficking.
Reported by Guimaraes et al. [83]	Ctx-DTA	Retrovirus-based insertional mutagenesis Genome-wide Loss-of-function Human haploid KBM7 cells	In the absence of ST3GAL5, 0-series ganglioside synthesis is turned on in a cell cycle-dependent manner.
Reported by Gilbert et al. [26]	Ctx-DTA	CRISPRi and CRISPRa Genome-wide Loss- and gain-of-function Human K562 cells	Upregulation or downregulation of one branch of ganglioside synthesis causes dynamic shifting in other branches.
Reported by Morgens et al. [97]	ricin	CRISPR-Cas9 Genome-wide Loss-of-function Human K562 cells	Known genes involved in N-linked glycosylation and fucosylation pathways.
Reported by Bassik et al. [98]	ricin	RNAi (shRNA) Genome-wide single shRNA for the primary screen, paired shRNA for the secondary screen Loss-of-function Human K562 cells	Known genes involved in membrane vesicle trafficking. C4orf41, KIAA1012, and C5orf44: as TRAPP interactors and defines two types of TRAPP complexes. RPS25: interacts with transcription factors ILF2 and ILF3. WDR11 and C17orf75: form a complex that may regulate ricin degradation through the autophagy pathway.
Reported by Tafesse et al. [152]	EtA	Retrovirus-based insertional mutagenesis Genome-wide Loss-of-function Human haploid KBM7 cells	Known genes involved in diphthamide biosynthesis. Known genes involved in membrane vesicle trafficking. GPR107: is a TGN-localized GPCR and is critical for retrograde transport.
Reported by Moreau et al. [99]	Ricin EtA	RNAi (siRNA) Genome-wide Loss-of-function Human HeLa cells	Known genes involved in membrane vesicle trafficking. Shared and unique trafficking pathways for different toxins and membrane-bound compartments.
Reported by Yamaji et al. [132]	SubAB	CRISPR-Cas9 Genome-wide Loss-of-function Human HeLa cells	Known genes involved in N-linked glycosylation. Known genes involved in O-linked glycosylation. Known genes involved in membrane vesicle trafficking. UNC50 and TMEM165. SLC39A9: acts as a Golgi Zn2+ transporter and globally regulates glycosylation. KDELR1/2 and JTB: involve in trafficking pathway.
Reported by Chang et al. [153]	typhoid toxin	CRISPR-Cas9 Genome-wide Loss-of-function Human HEK293T cells	Known genes involved in membrane vesicle trafficking. Known genes involved in ERAD. UNC50 and GPR107. TMED2: acts as a cargo receptor for Golgi-ER trafficking.
Reported by Anwar et al. [155]	Anthrax toxin	RNAi (siRNA) 1500 regulatory, trafficking, and cell-surface proteins Loss-of-function Human RPE1 cells	TMED10: is essential for anthrax toxin oligomerization on the cell surface. TMED2/10 complex facilitates the non-vesicular transfer of cholesterol and ceramide at the ER-Golgi membrane contact site, thereby controlling the formation of lipid nanodomains.
Reported by Carette et al. [18]	DT LFN-DTA	Retrovirus-based insertional mutagenesis Genome-wide Loss-of-function Human haploid KBM7 cells	Known genes involved in diphthamide biosynthesis. WDR85: is the ortholog of yeast YBR246W and involves diphthamide biosynthesis (later renamed as DPH7).
Reported by Tao et al. [113,154]	TcdB	CRISPR-Cas9 Genome-wide Loss-of-function Human HeLa cells	CSPG4 and FZDs: toxin receptors. EMC: facilities membrane protein biosynthesis. The EMC-dependent proteins have polar/charged residues within their transmembrane domains.
Reported by Tao et al. [33]	TcdA	CRISPR-Cas9 Genome-wide Loss-of-function Human HeLa cells	Known genes involved in sGAGs biosynthesis pathway. LDLR: facilitates toxin uptake. TMEM165.
Reported by Zhou et al. [117]	TcnA	CRISPR-Cas9 Genome-wide Loss-of-function Human HeLa cells	Known genes involved in sGAGs biosynthesis pathway. LDLR: facilitates toxin uptake.
Reported by Tian et al. [35]	TcdB4	CRISPR-Cas9 Genome-wide Loss-of-function Human HeLa cells	TFPI: toxin receptor. Known genes involved in the biosynthesis of GPI anchor.
Reported by Luo et al. [114]	TcdB4	CRISPR-Cas9 Genome-wide Loss-of-function Human HeLa cells	TFPI: toxin receptor. Known genes involved in the biosynthesis of GPI anchor.
Reported by Tromp et al. [134]	PVL HlgCB	CRISPR-Cas9 Genome-wide Loss-of-function Human U937 cells	Known genes involved in tyrosine sulfation. Known genes involved in sialyation. Sialyation-deficient cells have lower surface expression level of GPCRs.
Reported by Jeon et al. [156]	PVL	CRISPR-Cas9 Genome-wide Loss-of-function Human THP1 macrophages	FBXO11: regulates the inflammasome pathway and the expression of IL-1β through transcriptional regulation.
Reported by Drabavicius et al. [138]	ILY	CRISPR-Cas9 Genome-wide Loss-of-function Human near-haploid HAP1 cells	Known genes involved in the biosynthesis of GPI anchor. Known genes involved in N-linked glycosylation. Known genes involved in GSLs biosynthesis. Known genes involved in sGAGs biosynthesis pathway. Known genes involved in nucleotide sugar metabolism. TM9SF2.
Reported by Shahi et al. [141]	ILY	CRISPR-Cas9 Genome-wide Loss-of-function Human HeLa cells	Known genes involved in the biosynthesis of GPI anchor.
Reported by Virreira Winter et al. [149]	αHL	CRISPR-Cas9 Genome-wide Loss-of-function Human U937 cells	SYS1, ARFRP1, and TSPAN14: regulate the surface expression of toxin receptor ADAM10.
Reported by Song et al. [144]	Tc toxins	CRISPR-Cas9 Genome-wide Loss-of-function Human HeLa cells	Known genes involved in the biosynthesis of GPI anchor. Known genes involved in N-linked glycosylation. Known genes involved in sGAGs biosynthesis pathway. TMEM165 and TM9SF2.

* Screen strategies encompass various technical considerations, such as screen scale (partial genome or genome-wide), cell model, methods of generating mutations, screen format (loss-of-function or gain-of-function), and some special settings. ** The loss-of-function approach involved using a library targeting the human genome in Vero cells derived from green monkeys. *** The EHEC strain used in this screen expresses both T3SS and Stx.

## 7. Perspectives

The journey of studying bacterial toxins began in the 19th century. In 1884, Robert Koch pointed out that *Vibrio cholerae* induces disease through a secreted “poison,” which was later known as Ctx after more than half a century. In 1883, *Corynebacterium diphtheriae* was identified as the causative agent of diphtheria. In 1888, Émile Roux and Alexandre Yersin discovered and defined the first bacterial toxin, diphtheria toxin (DT), from the supernatant of *Corynebacterium diphtheriae* culture. This discovery demonstrated that bacteria could produce a particular substance that acts as the disease-causing agent, and the substance was named as bacterial toxin. In 1901, a toxin biologist, Emil von Behring, was awarded the first Nobel Prize in Physiology or Medicine for his work on developing antiserum against DT and tetanus toxin. Since then, more than 300 toxins have been discovered [157].

Significant progress has been made in terms of research on toxin biology and toxin–host interactions. On the toxin side, the biochemical and structural features of toxin have been elucidated, while on the host side, the cellular mechanism of their actions has been disclosed. These achievements have provided insights into how these potential bioterrorism agents have become the most poisonous substances in the world, with BoNT and TeNT being claimed as the most potent toxins, ranked No. 1 and No. 2, respectively [158]. On the other hand, these naturally evolved properties of toxins provide a powerful toolbox for engineering and developing toxin-based platforms. These platforms can be tailored to target specific cells or tissues, enable efficient intracellular delivery [159], and modulate signaling pathways.

This review specifically focuses on the combination of toxin biology and recently advanced genetic screen technologies—toxins serve as specific and robust probes, while genetic screens provide systematic and unbiased surveys. Therefore, toxin-based genetic screens can reveal a comprehensive list of host factors involved in toxin intoxication. Particularly, host factors that indirectly interact with toxins may play important roles in fundamental biological pathways and eventually benefit toxins. These host factors may be involved in controlling the biosynthesis of toxin receptors, facilitating toxin trafficking among membrane-bound organelles, or being required for the toxin’s enzymatic activity.

The successful identification of these factors highly relies on the employed screen strategies. For example, RNAi-based screens for ricin identified many trafficking factors, whereas CRISPR-based screens for ricin additionally revealed *N*-linked glycosylation and fucosylation pathways. Similarly, multiple CRISPR-based screens for Stx consistently identified *LAPTM4A* and *TM9SF2*, which had not been recognized by previous RNAi-based screens. Furthermore, the PVL screen on a regular cell line only provided information on the receptor-related pathways, whereas the same screen conducted in macrophages discovered regulators in the inflammasome pathway. Moreover, some minor conditions, such as toxin concentration and treatment time, can also have an impact. Future investigations in this area are required, involving more screens using various strategies and delving into the molecular mechanisms of these newly identified factors and pathways.

## Figures and Tables

**Figure 1 bioengineering-10-00884-f001:**
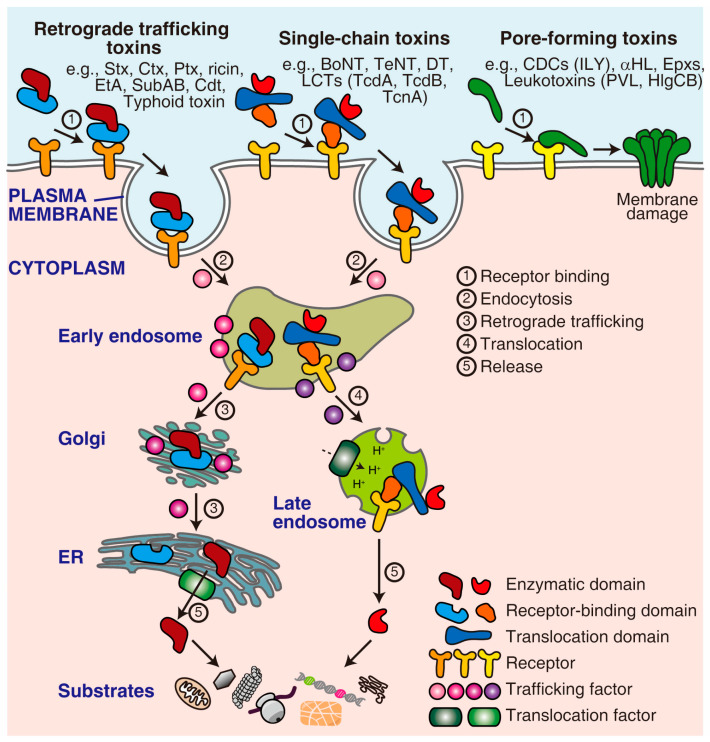
The mechanism of action of representative toxins. Pore-forming toxins directly act on the plasma membrane. In contrast, retrograde trafficking toxins and single-chain toxins need to enter the cell through a series of processes, including receptor binding, endocytosis, trafficking/translocation, release/activation, and eventually acting on their cytoplasmic substrates. This process requires multiple host factors, such as receptors, trafficking factors, and translocation factors. ER, endoplasmic reticulum; Stx, Shiga toxin; Ctx, cholera toxin; Ptx, pertussis toxin; EtA, *Pseudomonas aeruginosa* exotoxin A; SubAB, *Escherichia coli* subtilase cytotoxin; Cdt, cytolethal distending toxin; BoNT, botulinum neurotoxin; TeNT, tetanus neurotoxin; DT, diphtheria toxin; LCTs, large clostridial toxins; TcdA, *Clostridioides difficile* toxin A; TcdB, *Clostridioides difficile* toxin B; TcnA, *Clostridium novyi* alpha-toxin; CDCs, cholesterol-dependent cytolysins; ILY, *Streptococcus intermedius* intermedilysin; αHL, *Staphylococcus aureus* α-hemolysin; Epxs, *Enterococcus* pore-forming toxins; PVL, *Staphylococcus aureus* Panton-Valentine leucocidin; HlgCB, *Staphylococcus aureus* γ-haemolysin CB.

**Figure 2 bioengineering-10-00884-f002:**
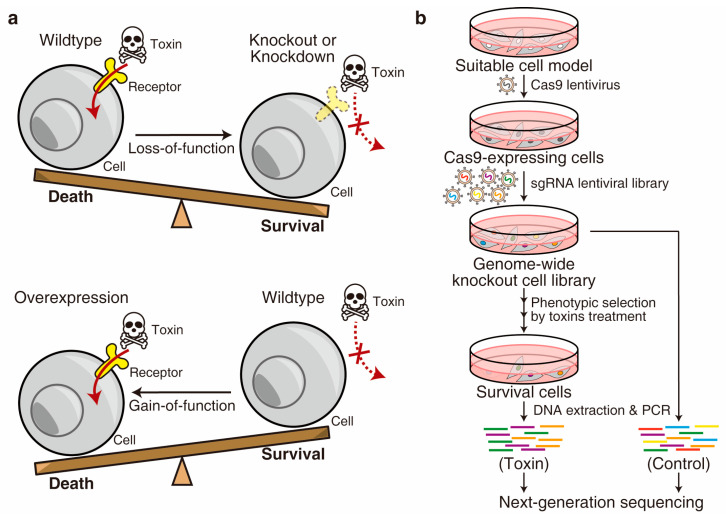
Forward genetic screen. (**a**) The screens can be conducted either in a loss-of-function or gain-of-function manner. (**b**) Schematic diagram of a representative genome-wide CRISPR-Cas9-mediated loss-of-function screen for a toxin.

**Figure 3 bioengineering-10-00884-f003:**
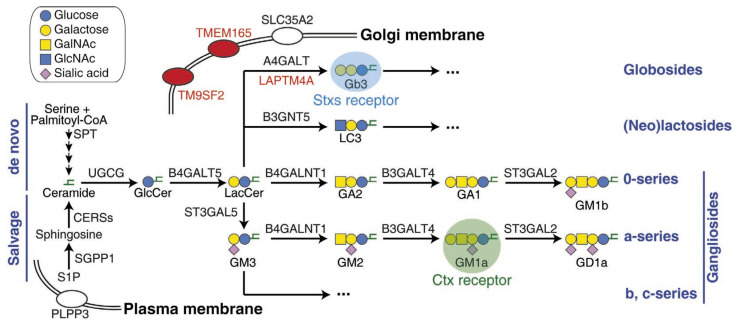
Scheme of GSL’s biosynthesis pathway and related factors. The abbreviations of GSLs recommended by IUPAC [40] are used in this figure. It highlights the Stxs receptor Gb3, Ctx receptor GM1, and newly identified factors through recent CRISPR screens. GalNAc, *N*-acetylgalactosamine; GlcNAc, *N*-acetylglucosamine.

**Figure 4 bioengineering-10-00884-f004:**
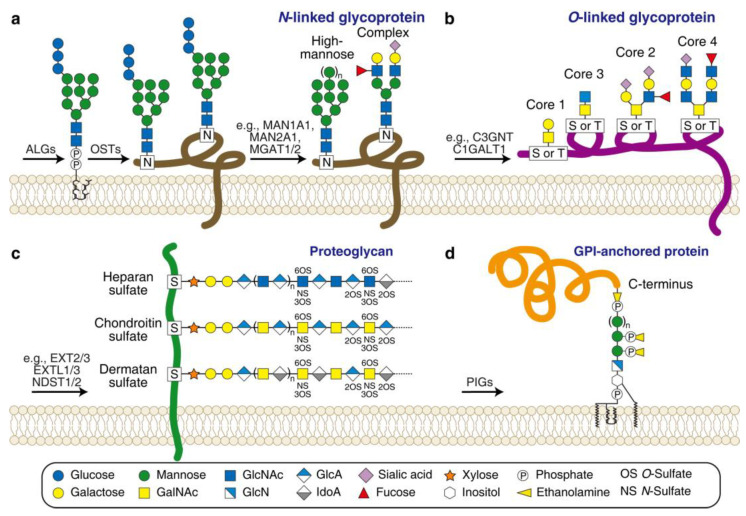
Scheme of the major types of protein glycosylation. (**a**) *N*-linked glycoprotein. ALGs (asparagine-linked glycosylation enzymes) are required for producing the dolichol-linked precursor oligosaccharide; OSTs (oligosaccharyltransferase complex subunits) are required for transferring the precursor oligosaccharide to an asparagine residue (marked as N) of a protein; a series of glycosidases and glycosyltransferases (e.g., MAN1A2/2A1 and MGAT1/2) are required for converting the precursor oligosaccharide to high-mannose-type or complex-type glycans. (**b**) *O*-linked glycoprotein. Four dominant core structures (Core 1to 4) are linked to a serine or threonine residue (marked as S or T) of a protein by a series of glycosyltransferases (e.g., C3GNT and C1GALT1). (**c**) Proteoglycan. Three representative sGAGs are linked to a serine residue (marked as S) of a core protein by a series of glycosyltransferases and sulfotransferases (e.g., EXT2/3, EXTL1/3, and NDST1/2). (**d**) GPI-anchored protein. PIGs (phosphatidylinositol glycan enzymes) are required for producing and ligating a GPI anchor to the C-terminus of a protein. Carbohydrate legends are shown at the bottom. GlcN, glucosamine; GlcA, glucuronic acid, IdoA, iduronic acid.

## Data Availability

Not applicable.

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
