# Peer review of "Gaining New Insights into Fundamental Biological Pathways by Bacterial Toxin-Based Genetic Screens"

_bioengineering, 2023, doi:10.3390/bioengineering10080884_

Round 1
Reviewer 1 Report
The manuscript addressed the topic (Using Bacterial Toxin-Based Genetic Screens for Discovering More About Crucial Metabolic Pathways). The manuscript is well-written and designed appropriately. However, the following points should be considered by the authors before accepting the manuscrip for publication.
1- Adding a brief history about the toxins discovery and Toxin-based genetic screen strategies.
2- Insert appropriate citation after each statments drived for other researchers works.
3- Add a advantage/disadvantage for each toxin-based genetic screen strategy (whenever possible).
4- Add a list of all abbreviations used in the manuscript.
5- Scientific names should be italic in the references

English language is generally OK. Minor changes need to be done.
Author Response
Response to Reviewers (Manuscript ID: bioengineering-2426477)
We thank the reviewers for their time and insightful suggestions. We are grateful for the support of the Editorial Office at Bioengineering for kindly giving us this opportunity to revise our manuscript. We have revised the manuscript to address each of the reviewers’ questions/concerns as outlined below. All the revisions to the manuscript have been highlighted. These changes further improved our manuscript and we appreciate it very much for reviewers’ and editor’s consideration.
Reviewer #1:
Q1. Adding a brief history about the toxins discovery and Toxin-based genetic screen strategies.
Response: Toxin discovery has a long history. It would require a significant amount of space to introduce the story of every toxin mentioned in this review. Moreover, the primary focus of this review is not to delve into the individual toxins themselves, but rather to explore the utilization of toxins as probes (base on the current knowledge on toxin biology) to investigate the host cell pathways. Therefore, we have only briefly mentioned the early studies on cholera toxin and diphtheria toxin in the Perspectives section, highlighting their significance in the history of toxin research. In response to the reviewer’s suggestion, we have expanded the introduction to include more historical context on the discovery of diphtheria, as a representative of the concept of bacterial toxins. We also add more introduction on the history of toxin-based genetic screen strategies accordingly.
Q2. Insert appropriate citation after each statments drived for other researchers works.
Response: As the reviewer suggested, more citations have been inserted.
Q3. Add a advantage/disadvantage for each toxin-based genetic screen strategy (whenever possible).
Response: Following reviewer’s suggestion, a paragraph has been added to summarize the advantages and disadvantages of each screen strategies mentioned in this review.
Q4. Add a list of all abbreviations used in the manuscript.
Response: We have reinstated the abbreviation list that was previously deleted due to manuscript length considerations. The list is included below, preceding the References section.
Q5. Scientific names should be italic in the references
Response: The references were generated automatically by EndNote software (version X8) following the MDPI citation style (refer to https://www.mdpi.com/authors/references, and https://endnote.com/style_download/mdpi/).
Reviewer 2 Report
It is an interesting review article, which can be considered due to its importance in the field. I have gone through the manuscript. It is well-written, thorough and in-depth. However, there are some places which needs further minor corrections.
Firstly, minor but thorough English editing is required. Please revise the manuscript taking help from a colleague who is proficient in English and familiar with the subject matter, who can review your manuscript. Numerous typo and grammatical errors are there, which can easily be rectified.
The whole manuscript relies on Genetic screen technology and Genetic screens. However, authors must understand that it is a review article and this term is not very common. Authors must describe in the introduction that what is genetic screens in detail, what is its significance? and what are the uses? This will give a good start to the review and any reader will be able to understand.
Similarly, authors have particularly focussed on 1. retrograde trafficking toxins, 2. single-chain toxins and 3. pore-forming toxins. Before authors go further, they must explain these toxins, their relevance, their involvement in multiple pathways and possible mechanism, before directly jumping into Factors Required for Biosynthesis of Glycosphingolipids. This will maintain the reading continuity.
In table 1, I suggest screens column should be the last column.
In table 1, I cannot see any host factors instead only functions.
Similarly, in table 1, column screen strategy is not strategy. I think authors must choose appropriate headings for each column. There is no intended meaning of the headings matching with what is portrayed in the table.
Overall, the review is good and considerable.
Minor typos mistakes.
Author Response
Response to Reviewers (Manuscript ID: bioengineering-2426477)
We thank the reviewers for their time and insightful suggestions. We are grateful for the support of the Editorial Office at Bioengineering for kindly giving us this opportunity to revise our manuscript. We have revised the manuscript to address each of the reviewers’ questions/concerns as outlined below. All the revisions to the manuscript have been highlighted. These changes further improved our manuscript and we appreciate it very much for reviewers’ and editor’s consideration.
Reviewer #2:
Q1. Firstly, minor but thorough English editing is required. Please revise the manuscript taking help from a colleague who is proficient in English and familiar with the subject matter, who can review your manuscript. Numerous typo and grammatical errors are there, which can easily be rectified.
Response: We thank the reviewer for this suggestion. The manuscript has undergone multiple rounds of review by native English speakers, resulting in the correction of typos and grammatical errors. In order to continue progressing the manuscript, we kindly request that any additional errors that are identified be brought to our attention.
Q2. The whole manuscript relies on Genetic screen technology and Genetic screens. However, authors must understand that it is a review article and this term is not very common. Authors must describe in the introduction that what is genetic screens in detail, what is its significance? and what are the uses? This will give a good start to the review and any reader will be able to understand.
Response: We agree with the reviewer and we have added the introduction for genetic screens accordingly in accordance with their feedback.
Q3. Similarly, authors have particularly focussed on 1. retrograde trafficking toxins, 2. single-chain toxins and 3. pore-forming toxins. Before authors go further, they must explain these toxins, their relevance, their involvement in multiple pathways and possible mechanism, before directly jumping into Factors Required for Biosynthesis of Glycosphingolipids. This will maintain the reading continuity.
Response: We understand and appreciate the reviewer's concern regarding the introduction of toxins themselves. To ensure a smooth reading experience, we have structured the body of this review into distinct sections based on biological pathways rather than individual toxins. Within each section, we have provided the necessary background and biology of each toxin, in terms of “relevance, involvement in multiple pathways and possible mechanism” as suggested by the reviewer. We already have included Figure 1 and its legends at the beginning of the review to provide an overview and avoid repetitive introductions of these aspects throughout the text. While we acknowledge the value of introducing toxins in more detail, we believe that the comprehensive list of references at the beginning that recapitulate these details, allow interested readers to access relevant resources for a deeper understanding. Hopefully this approach could maintain the flow of the review while also address the reviewer's concerns.
Q4. In table 1, I suggest screens column should be the last column.
Response: We keep describing each screen by using the main authors’ names in the text, so we listed the “Screens” as the first column, which is easier for the reader to track. To be clearer, we have modified this column as “Reported by (author’s name) [ref#]”, and we believe that this modification will enhance clarity and reduce confusion when referring to specific screens in the table.
Q5. In table 1, I cannot see any host factors instead only functions.
Response: The last column of the table, titled “Identified Host Factors and Functions”, includes newly identified host factors along with their functions. However, certain host factors were not listed for the following reasons: (1) Some of these host factors have been identified in multiple screens and their functions have already been introduced in the first screen. To maintain the conciseness of the table, we have omitted the functions of these host factors. (2) Certain host factors are among the top hits in the screens, but their functions have not been thoroughly characterized. The reviewer's comment about “not seeing any host factors” seems to refer to the absence of the known genes in established pathways. We did not specifically mention these known genes (actually they have been discussed in the review text) because they and their functions had been reported prior to the toxin-based screens. The purpose of our review was to highlight the authentic host factors involved in the biological process rather than reiterate the known genes with established functions. Consequently, the newly identified hits with unknown functions hold significant potential for further study. While it is possible to include them in the table, doing so may make the table overly cumbersome, thus hindering the emphasis on the newly identified host factors.
Q6. Similarly, in table 1, column screen strategy is not strategy. I think authors must choose appropriate headings for each column. There is no intended meaning of the headings matching with what is portrayed in the table.
Response: We have made sure that the term “screen strategy” includes the detailed screen methods, such as screen scale (partial genome or genome-wide), cell model, the way to generate mutation, loss- or gain-of-function, some special settings, and other considerations in the genetic screen field. To be more precise, we added a footer as description after the table. We welcome the reviewer to specify any content that may be missing or not in accordance with the heading "screen strategy," and we will be glad to address any concerns accordingly.
Reviewer 3 Report
This review manuscript focused on recent toxin-based genetic screenings such as the CRSPER-cas9 system and described that those screenings identified new players involved in and provided new insights into the fundamental biological pathways. The review is interesting and could offer an understanding of the biological mechanism based on intoxication and stimulate new therapeutic approaches for infectious diseases and genetic disorders with defects in these players and pathways. However, the manuscript has some minor problems.
1. The authors summarized the toxin-based genetic screenings in Table 1. But this table is very complicated to understand. The authors should recompile the table around screening strategies or toxins.
2. Line 603, “know” should replace “known.”
Author Response
Response to Reviewers (Manuscript ID: bioengineering-2426477)
We thank the reviewers for their time and insightful suggestions. We are grateful for the support of the Editorial Office at Bioengineering for kindly giving us this opportunity to revise our manuscript. We have revised the manuscript to address each of the reviewers’ questions/concerns as outlined below. All the revisions to the manuscript have been highlighted. These changes further improved our manuscript and we appreciate it very much for reviewers’ and editor’s consideration.
Reviewer #3:
Q1. The authors summarized the toxin-based genetic screenings in Table 1. But this table is very complicated to understand. The authors should recompile the table around screening strategies or toxins.
Response: We appreciate the reviewer’s suggestion. We listed the “Screens” as the first column because we keep describing each screen by the authors’ name in the text, making it easier for readers to track and associate the information. We have listed the screens in a specific order based on the types of screening toxins, specifically arranging them as retrograde trafficking toxins, single-chain toxins, pore-forming toxins, and others. However, we did not recompile the table because each group of researchers identified pathways that may have overlaps but are not completely identical. We wanted to avoid any confusion regarding their specific findings and ensure clarity in presenting the information as originally reported by each research group.
Q2. Line 603, “know” should replace “known.”
Response: Thanks for pointing it out and it has been corrected.
Round 2
Reviewer 2 Report
Manuscript can be accepted in its current form.
Acceptable